# Analysis of the Causes of Solitary Pulmonary Nodule Misdiagnosed as Lung Cancer by Using Artificial Intelligence: A Retrospective Study at a Single Center

**DOI:** 10.3390/diagnostics12092218

**Published:** 2022-09-13

**Authors:** Xiong-Ying Wu, Fan Ding, Kun Li, Wen-Cai Huang, Yong Zhang, Jian Zhu

**Affiliations:** 1Department of Otolaryngology-Head and Neck Surgery, General Hospital of Central Theater Command of the People’s Liberation Army, Wuhan 430070, China; 2Department of Orthopaedics, General Hospital of Central Theater Command of the People’s Liberation Army, Wuhan 430070, China; 3Department of Anesthesiology, General Hospital of Central Theater Command of the People’s Liberation Army, Wuhan 430070, China; 4Department of Radiology, General Hospital of Central Theater Command of the People’s Liberation Army, Wuhan 430070, China; 5Department of Integrative Medicine, General Hospital of Central Theater Command of the People’s Liberation Army, Wuhan 430070, China; 6Department of Thoracic Cardiovascular Surgery, General Hospital of Central Theater Command of the People’s Liberation Army, Wuhan 430070, China

**Keywords:** solitary pulmonary nodule, artificial intelligence, coronavirus disease 2019, convolutional neural networks, deep learning, lung cancer

## Abstract

Artificial intelligence (AI) adopting deep learning technology has been widely used in the med-ical imaging domain in recent years. It realized the automatic judgment of benign and malig-nant solitary pulmonary nodules (SPNs) and even replaced the work of doctors to some extent. However, misdiagnoses can occur in certain cases. Only by determining the causes can AI play a larger role. A total of 21 Coronavirus disease 2019 (COVID-19) patients were diagnosed with SPN by CT imaging. Their Clinical data, including general condition, imaging features, AI re-ports, and outcomes were included in this retrospective study. Although they were confirmed COVID-19 by testing reverse transcription-polymerase chain reaction (RT-PCR) with severe acute respiratory syndrome coronavirus 2 (SARS-CoV-2), their CT imaging data were misjudged by AI to be high-risk nodules for lung cancer. Imaging characteristics included burr sign (76.2%), lobulated sign (61.9%), pleural indentation (42.9%), smooth edges (23.8%), and cavity (14.3%). The accuracy of AI was different from that of radiologists in judging the nature of be-nign SPNs (*p* < 0.001, κ = 0.036 < 0.4, means the two diagnosis methods poor fit). COVID-19 patients with SPN might have been misdiagnosed using the AI system, suggesting that the AI system needs to be further optimized, especially in the event of a new disease outbreak.

## 1. Introduction

The solitary pulmonary nodule (SPN) refers to a single round lesion in the lung with a maximum diameter of ≤30 mm, which is surrounded by lung tissue and without atelectasis, obstructive pneumonia, hilar lymphadenopathy, and pleural effusion [1]. On the one hand, according to the density of the nodules, they can be divided into pure ground-glass nodules (pGGNs) and mixed ground-glass nodules (mGGNs) [2]. On the other hand, according to the nature of the nodules, they can be divided into benign and malignant nodules, but they are more common in lung cancer patients. With the advancement of science and technology and the improvement of people’s health awareness, an increasing number of isolated lung nodules have been discovered [3]. Computed tomography (CT) is currently one of the first choices for screening lung nodules. With the development of CT scanning technology, the thickness of the scanning layer has become thinner, and smaller SPNs have been found in CT images.

However, a set of data of a patient contains a large number of CT images, which leads to an inevitable problem in manual screening: The process of finding pulmonary nodules from these images is time-consuming and laborious. The coronavirus disease 2019 (COVID-19) epidemic, in particular, is a serious threat to global public health security that has brought huge challenges and a huge workload to the management of medical staff [4]. Long-term labor causes fatigue in doctors and reduces the effectiveness of screening, leading to a missed diagnosis of SPNs [5]. In recent years, with the development of artificial intelligence (AI) technology, computer-assisted screening can reduce the workload of doctors and reduce the missed diagnosis rate by automatically locating lesions [6]. By using this auxiliary diagnosis, doctors can also improve diagnosis efficiency.

AI can provide quantitative data to assist in clinical decision making. When the diagnosis is unclear or divergent, it can automatically distinguish between benign and malignant lesions based on digital image features. At present, AI is mainly based on deep learning to realize candidate detection and false-positive reduction [7,8]. Through convolutional neural networks (CNNs) with representation learning capabilities, which are one of the most widely used deep learning models, AI can realize high-precision judgments of benign or malignant SPNs [9]. With the help of AI technology, the accuracy of SPN classification was 92.0%, the sensitivity was 93.6%, and the specificity was 39.3% [10].

This report presents a study of patients with SPNs hospitalized at the General Hospital of the Central Theater Command of the People’s Liberation Army during the COVID-19 epidemic, as well as an evaluation of the diagnostic accuracy of AI on SPNs at the time of the emergence of this new disease.

## 2. Methods

### 2.1. Patient Inclusion/Exclusion Criteria

We retrospectively analyzed the clinical data of those patients who were diagnosed with SPNs for the first time through chest CT examinations from 22 January 2020 to 15 August 2020. All the included patients had postoperative pathological results or positive RT-PCR results for COVID-19 within one month. The pulmonary nodules disappeared in all of these patients during follow-up, and no pathological examination results were available for any of them. Deep learning algorithms were used to extract features of all the nodules and predict whether the nodules were malignant using the established convolutional neural network model. In addition, two senior diagnostic radiologists independently analyzed the CT images of the detected SPNs through a blinded method to determine the nature of all SPNs. In the event of a difference of opinion, the chief diagnostic chest imaging physician was asked to lead a discussion to reach a final agreement. The results of CT examination reports were written by these senior diagnostic radiologists who were legally responsible. The CT analysis also included the distribution of the SPNs, the location of the SPNs, the characteristics of the SPNs, and external involvement. All SPNs were confirmed by pathological examinations or followed up with clinical treatment.

### 2.2. CT Examination and AI Analysis

The CT scan was performed in a specific computer room with a spiral scan using a designated Toshiba 16-slice CT. The scanning was carried out in breath-holding mode, applying automatic tube current modulation, ranging from the tip of the lung to the bottom of the lung. The voltage of the tube was 120 kV, and the thickness and spacing of the scanning layer were 0.5–2 mm. The tube current was adjusted to ensure that the CTDIvol value was 7 mGy. The pitch was 1.3, and the pedal scan direction was also used. The environment and equipment in the computer room were completely disinfected because COVID-19 was spreading in Wuhan. The technicians had to take protective measures and carry out hand disinfection and cleaning immediately after each contact with the patient. All patient waste had to be disposed of following the infectious clinical waste process. The AI software supported by Hangzhou Yitu Medical Technology Limited Company was used to assist with diagnosis.

### 2.3. Statistical Analysis

The SPSS 23.0 statistical software was used to analyze all the data. According to the results of SPN prediction, the accuracy, sensitivity, and specificity of AI technology and the radiologists’ diagnosis were calculated. The accuracy was expressed by the ratio of predicting the correct total number of SPNs to the number of summarized SPNs. The sensitivity was expressed by predicting the correct ratio of the total number of malignant SPNs to the total number of malignant SPNs. The specificity was expressed by predicting the correct ratio of the total number of benign SPNs to the total number of benign SPNs. The measurement data with a normal distribution are described as the mean ± standard deviation (X ± S), while the measurement data with a non-normal distribution are described as the median (IQR). The counting data are described as examples and percentages (n (%)). The chi-square test was used to compare various factors between the groups. The difference was statistically significant at *p* < 0.05.

## 3. Results

### 3.1. AI Prediction Results

A total of 61 patients were enrolled in this study, 21 of which had confirmed COVID-19 through throat-swab RT-PCR. The pulmonary nodules disappeared in all of these patients during follow-up, and no pathological examination results were available in any of them. The remaining 40 patients were diagnosed with SPNs through postoperative pathology, including 34 malignant SPNs and 6 benign SPNs. Therefore, the final diagnosis was 34 malignant SPNs and 27 benign SPNs (including 21 benign SPNs of COVID-19 and 6 benign SPNs of tumors). Among all the 34 malignant SPNs, AI successfully predicted 31 SPNs (91.2%) as malignant nodules, and the remaining 3 SPNs (8.8%) were misdiagnosed as benign nodules. Among all the 27 benign SPNs, AI successfully predicted that 5 SPNs (18.5%) were benign nodules, and the remaining 22 SPNs (81.5%) were misdiagnosed as malignant nodules. The accuracy rate of the diagnosis using AI technology was 59.0%, the sensitivity was 91.2%, the specificity was 18.5%, and the false-positive rate was 81.5% (Table 1).

### 3.2. Radiologists’ Prediction Results

Among all the 34 malignant SPNs, the radiologists successfully predicted 32 SPNs (94.1%) as malignant nodules, and the remaining 2 SPNs (5.9%) were misdiagnosed as benign nodules. Among all the 27 benign SPNs, the radiologists successfully predicted that 25 SPNs (92.6%) were benign nodules, and the remaining 2 SPNs (7.4%) were misdiagnosed as malignant nodules. The accuracy rate of the diagnosis of these radiologists was 93.4%, the sensitivity was 94.1%, the specificity was 92.6%, and the false-positive rate was 7.4% (Table 1).

### 3.3. Comparison between AI Prediction Results and Radiologist Prediction Results

The accuracy and specificity of the radiologists in predicting the nature of SPNs were higher than those of the AI system. However, the AI system had a high false-positive rate, and its sensitivity was almost the same as that of the radiologists (Table 2). Further investigation found that these differences were mostly caused by the outcome of benign SPNs. The accuracy of the AI system was different from that of the radiologists in judging the nature of benign SPNs (*p* < 0.001, Table 3). To identify the possible reasons for these large differences, we further tracked the clinical data of 27 patients with benign SPNs. It is a wonder that only six SPNs were traditionally benign nodules: one lipoma, one foreign body granuloma, two hamartoma, and two alveolar epithelial hyperplasia cases with interstitial inflammatory cell infiltration. The remaining 21 SPNs were COVID-19.

### 3.4. Clinical Data of Misdiagnosed benign SPNs with AI

Among the 21 cases, the average age was 41.71 ± 16.04 years (no children or adolescents were infected), including 9 males (42.9%) and 12 females (57.1%). Ten cases (47.6%) had a direct exposure history of close contact. Five patients (23.8%) had underlying diseases, including hypertension, diabetes, tuberculosis, and hypothyroidism. Eighteen patients (85.7%) had no history of smoking. None of the patients had a personal or family history of lung cancer.

On admission, the clinical symptoms were mainly fever in seven cases (33.3%, without high-fever patients), cough in six cases (28.6%), three of which were dry cough (14.3%), fatigue in six cases (28.6%), chills in six cases (28.6%), muscle soreness in four cases (19.0%), and sore throat in three cases (14.3%). The less common symptoms were headache in two cases (9.5%), rhinorrhea in two cases (9.5%), nausea in two cases (9.5%), and chest tightness in two cases (9.5%). It should be noted that five patients (23.8%) did not have any special symptoms at the time of onset, but no further progression of the lesion was found during hospitalization (Table 4). Fifteen patients (71.4%) eventually developed a fever. All 21 patients underwent chest CT re-examination in our hospital, and by comparing the re-examination of their chest CT with the first CT images, the outcome of their pulmonary treatment progress was evaluated.

The imaging characteristics of their chest CT showed ground-glass shadow, 16 cases of them (76.2%) showed burr signs, 13 cases (61.9%) showed lobulated signs, 9 cases (42.9%) showed pleural indentations, 5 cases (23.8%) showed smooth edges, and 3 cases (14.3%) showed cavities (Figure 1 and Figure 2). The density of lesions was usually heterogeneous; 13 of them (61.9%) showed pGGNs, and 8 cases (38.1%) showed mGGNs. The maximum diameter of the SPNs was 30 mm. There were 3 patients (14.3%) with an SPN less than 10 mm, 8 patients (38.1%) with an SPN of 10–20 mm, and 10 patients (47.6%) with an SPN more than 20 mm. The average volume of the SPNs was 3280 ± 2646 mm³. In the first CT examination, no patients had hollow lungs, hilar or mediastinal lymph nodes, pericardial effusion, pneumothorax, or atelectasis. In the reports of the AI system, 16 patients (76.2%) were considered high-risk patients for lung cancer, and 5 were intermediate-risk patients (23.8%). These results mean that intervention measures should be taken as soon as possible (Table 5).

## 4. Discussion

In recent years, AI technology has made great progress and has gradually been applied in the processing and analysis of medical images [11]. AI models automatically extract image features and apply machine learning algorithms to screen lesions and make diagnoses. Chest CT images have been able to distinguish lung adenocarcinoma, squamous cell carcinoma, small cell lung cancer, and other types based on the CNN algorithm model to classify lung cancer pathological images, which showed good application prospects [12]. Ciompi et al. also built a set of models that could classify the morphological characteristics of lung nodules based on CNN algorithms. They realized the classification of solid, subsolid, calcified, and non-solid nodules and obtained a higher accuracy rate [13]. All the signs support the good performance of AI in the detection and classification of pulmonary nodules.

However, in this special period of the COVID-19 outbreak, AI seems to have encountered new challenges. COVID-19 is a new respiratory infectious disease that was first reported in Wuhan, Hubei Province, China, in December 2019 [14]. The number of confirmed cases is still growing at a high speed every day [15]. In most of the existing reports, COVID-19 mainly manifests as bilateral lung involvement on imaging, with multiple small patchy shadows and interstitial changes in the early stage, especially in the peripheral zone [16,17]. Then, they develop into multiple ground-glass shadows or infiltration shadows in the bilateral lungs [18].

Accurately determining the nature of SPNs has always been a challenge for clinicians. An accurate diagnosis of COVID-19 was also challenging, especially in the early stages of the outbreak. If these patients only had SPNs based on CT findings or were asymptomatic, and COVID-19 was not excluded, the diagnosis was much more complicated.

Therefore, an accurate diagnosis is crucial in treatment selection and planning for each lung cancer or COVID-19 patient. It has been reported that an AI system can detect COVID-19 and other lung diseases that currently have a lower rate of contact transmission [19]. The sensitivity and accuracy of classifying SPNs by the AI system were better than those of the radiologists, as in some studies [20,21,22]. These high accuracy rates seemed to be related to COVID-19 patients with bilateral lung lesions, not to COVID-19 patients with SPNs [23]. The AI system could account for nodule size, volume, margins, attenuation, and other radiological factors consistently without requiring subjective judgment or data entry from the reading radiologist or pulmonologist [24]. The AI system plays a role in identifying low-risk nodules that do not need further surveillance and uses CT imaging features as the main basis for judging the nature of pulmonary nodules [25].

However, the AI system seemed to have a large error diagnosis in our study. We considered the following reasons. First, it might be that the CT manifestations of COVID-19 were diverse, most of which were multiple lesions in the bilateral lungs, and few were isolated lesions. SPNs were not the main imaging presentation. Second, when the CT manifestation of COVID-19 was an SPN, it also had pleural indentations, burr signs, and lobulated signs, such as lung cancer nodules. Third, the AI system had never learned the COVID-19 imaging presentation before because it is a new disease. Fourth, pathological examinations are ultimately the gold standard of diagnosis for lung cancer, while COVID-19 is confirmed with an RT-PCR test, as it has no way to obtain pathological specimens. The error of RT-PCR testing is larger than that of a pathological examination. Fifth, populations of all ages are generally susceptible to COVID-19, while lung cancer generally occurs in special populations with high-risk factors, such as advanced age and smoking [22]. The AI system could recognize these special signs, which would improve the accuracy of the diagnosis. Sixth, the classification of lung cancer is generally divided into small cell lung cancer and non-small cell lung cancer. However, the types of pneumonia can be divided into bacteria, fungal pneumonia, and viruses, including coronaviruses, parainfluenza viruses, and adenoviruses. Moreover, coronaviruses are divided into four genera: α, β, γ, and δ. SARS-CoV-2 is a new type of coronavirus in genus β [26]. For the AI system, the addition of new diverse types of COVID-19 was even more difficult to diagnose. Seventh, the AI software supported by Hangzhou Yitu Medical Technology Limited Company is usually used to diagnose lung cancer and may not apply to COVID-19.

In reality, it seems very difficult to realize the automatic judgment of benign and malignant lesions. It is impossible to reach the level of the clinician’s prediction at this stage. The first reason might be the complexity of benign and malignant nodules. There might be some characteristic differences between benign and malignant nodules. For example, calcification points were generally benign, while the larger nodules with burrs were generally malignant. However, the difference in these characteristics is not an absolute standard. Generally, only the results of pathological tests can be used as the basis for the final diagnosis. The second reason is the difficulty of obtaining the data. When predicting the nature of SPNs, doctors consider the relevant information of their patients. However, this information falls under patient privacy, thus involving data and information security and even ethical issues.

The training of CNN models usually requires a large number of standardized datasets to avoid overfitting, which is serious in the case of a small number of training samples and might be improved through transfer learning [27]. However, the acquisition, standardization, and security of medical image data were strict. Multi-center big data research in line with standards and quality control is relatively rare in China, especially for new diseases such as COVID-19. The function of CNNs is a black box, which makes it difficult to determine and explain how AI models draw conclusions [28]. In future research, we might have to perform more extensive research on the explanation of the internal mechanism of deep learning systems using AI technology.

To prevent misdiagnosis, we might need to consider the following measures during the COVID-19 epidemic: First, patients’ medical history has to be examined in detail, including the contact history of COVID-19 patients or whether they had a fever, cough, diarrhea, sore throat, or other symptoms in the past 2 weeks. Second, routine blood parameters, blood IgG and IgM analyses, and a nucleic acid examination of pharyngeal swab specimens need to be repeatedly tested. Third, instead of relying extensively on AI software images/CT images, we should focus on patients and their changing conditions. Fourth, as the incubation period of COVID-19 is approximately 2 weeks, we should not treat patients with lung cancer in merely 14 days unless we know their diagnosis [29]. The invasive pathologic examination of the tumor is also not recommended in such a short time. After all, lung cancer usually does not metastasize or spread during this short time.

The work presented here has limitations. First, this study is a single-center retrospective study with small sample size, and the analysis of our results may be biased. Second, this study only focused on the diagnosis of SPNs, which reduced the accuracy of the AI system’s diagnosis. Third, the AI system requires deep learning, but we analyzed these data at the beginning of the COVID-19 outbreak.

In summary, our study showed that patients with only SPNs may present with COVID-19 during the epidemic, which is easily misdiagnosed as lung cancer using AI technology. This suggests that the AI system needs to be further optimized, especially in the event of a new disease outbreak.

## Figures and Tables

**Figure 1 diagnostics-12-02218-f001:**
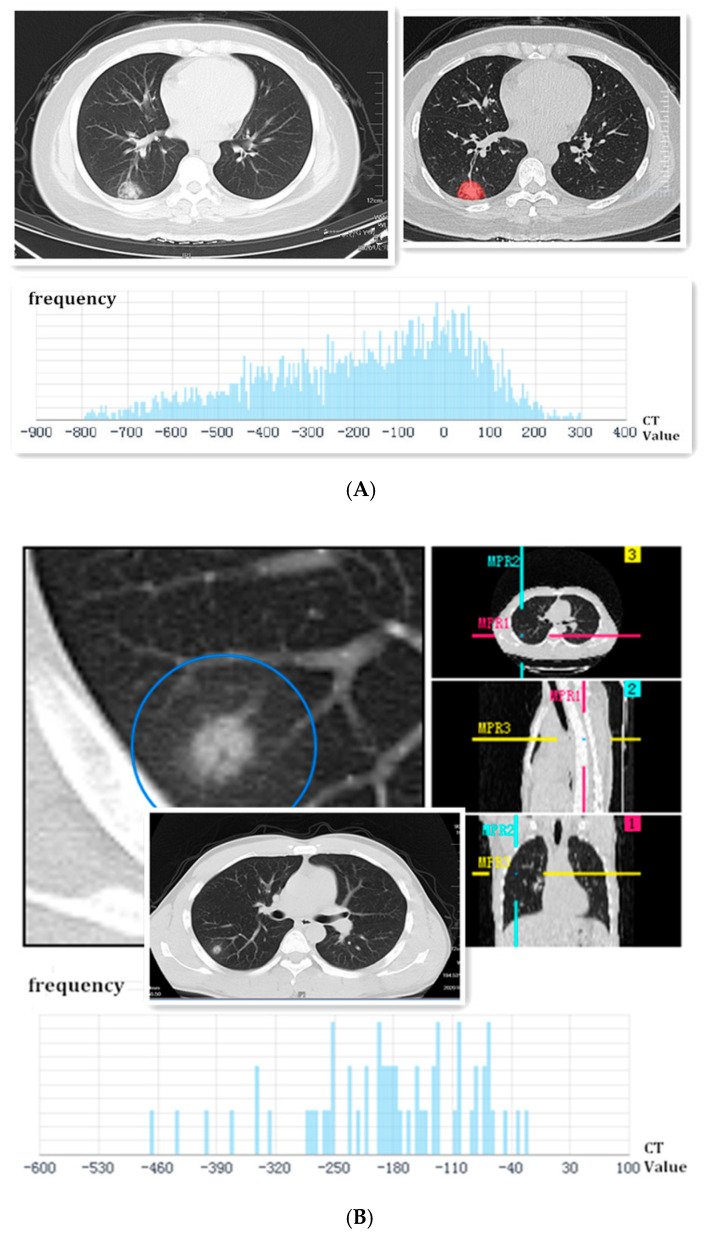
(**A**) Female, 37 years old, a coughing patient without sputum production and fever. The image first displayed a single ground-glass shadow, which had cavities and smooth edges in the peripheral area of the right lower lobe on admission. The SPN was marked by feature extraction of the AI system. Finally, it was identified as a high-risk nodule, and the frequency of CT values was analyzed using AI below the picture. This SPN was confirmed as COVID-19 via throat-swab RT-PCR; (**B**) male, a 33-year-old patient with a low fever. The image first displayed a localized consolidation shadow in the peripheral area of the right lower lobe on admission. The lobulated sign could be seen in the axial section, coronal section, and sagittal section. Finally, it was identified as a high-risk nodule, and the frequency of CT values was analyzed using AI below the picture. This SPN was confirmed to be COVID-19 through throat-swab RT-PCR.

**Figure 2 diagnostics-12-02218-f002:**
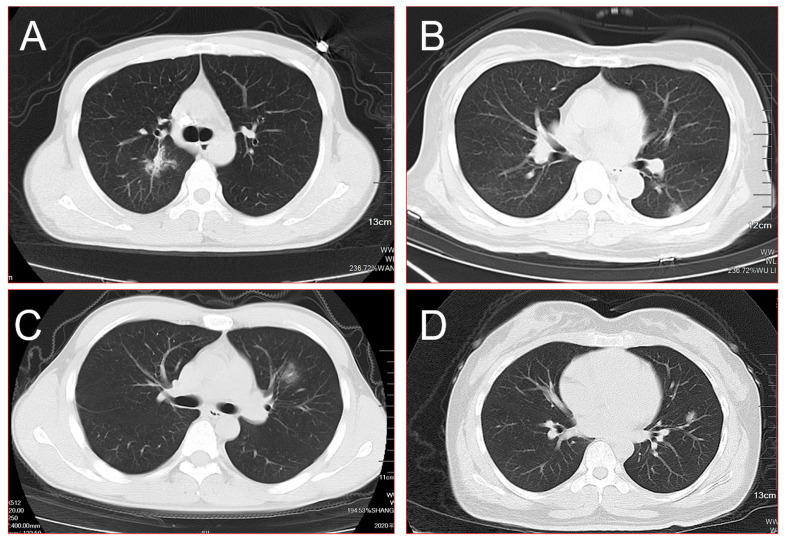
(**A**) Burr sign, lobulated sign, pleural indentation, and cavity could be seen in the central area of the right upper lobe simultaneously; (**B**) a single mixed ground-glass shadow was present in the peripheral area of the left lower lobe on admission; (**C**) a single ground-glass shadow with the blurred border was found in the left upper lobe; (**D**) a single mixed ground-glass shadow combined with cavities was found in the left upper lobe. All of these SPNs were finally identified as high-risk nodules using the AI system and were confirmed as COVID-19 through throat-swab RT-PCR.

**Table 1 diagnostics-12-02218-t001:** The prediction results of the 61 SPNs.

	AI Technology		Radiologist		
Pathology or RT-PCR Results	Correct	FALSE	Correct	FALSE	Total
Malignant	31	3	32	2	34
Benign	5	22	25	2	27

**Table 2 diagnostics-12-02218-t002:** Comparison between AI prediction results and radiologist prediction results.

	Accuracy Rate	Sensitivity	Specificity	False-Positive Rate
AI	59.00%	91.20%	18.50%	81.50%
Radiologist	93.40%	94.10%	92.60%	7.40%

**Table 3 diagnostics-12-02218-t003:** The prediction results of benign SPNs.

	AI Technology		*McNemar Test*	*Kappa*
Radiologist	Correct	FALSE		
Correct	5	20	<0.001	0.036
FALSE	0	2		

Κ < 0.4 means the two diagnosis methods fit poorly.

**Table 4 diagnostics-12-02218-t004:** Patient characteristics of COVID-19 patients with SPN.

	Patients (n = 21)
**Patients demographics**	
Mean age, years (range)	41.71 ± 16.04 (25–71)
Men	9 (42.9%)
Women	12 (57.1%)
**Exposure history**	
Exposure	10 (47.6%)
Unknown exposure	11 (52.4%)
**Current smoking**	3 (14.3%)
**Family history of cancer**	0
**Comorbid conditions**	
Any	5 (23.8%)
Hypertension	2 (9.5%)
Diabetes	2 (9.5%)
Tuberculosis	2 (9.5%)
Hypothyroidism	1 (4.8%)
**Signs and symptoms**	
Fever	7 (76%)
Cough	6 (28.6%)
Sputum production	3 (14.3%)
Fatigue	6 (28.6%)
Chills	6 (28%)
Muscle soreness	4 (28%)
Sore throat	3 (14.3%)
Headache	2 (9.5%)
Rhinorrhea	2 (9.5%)
Chest tightness	2 (9.5%)
Nausea	2 (9.5%)
Asymptomatic Patients	5 (23.8%)

**Table 5 diagnostics-12-02218-t005:** CT findings and AI results of COVID-19 patients with SPN.

	Patients (n = 21)
**Distribution**	
Periphery distribution	15 (71.4%)
Central distribution	6 (28.6%)
**Patterns of the SPN**	
Burr sign	16 (76.2%)
Lobulated sign	13 (61.9%)
Pleural indentation	9 (42.9%)
Smooth edges	5 (23.8%)
Cavity	3 (14.3%)
**Density of the SPN**	
Pure ground-glass nodule	13 (61.9%)
Mixed ground-glass nodule	8 (38.1%)
**Diameter of the SPN**	
<10 mm	3 (14.3%)
10 mm–20 mm	8 (38.1%)
>20 mm	10 (47.6%)
**AI results**	
High-risk nodules	16 (76.2%)
Medium-risk nodules	5 (23.8%)
**Progression of SPN**	
No development	5 (23.8%)
Develop to one side of lungs	4 (19.0%)
Develop to both sides of lungs	9 (42.9%)
Develops to all lobes of bilateral lungs	3 (14.3%)

## Data Availability

The datasets generated during the current study are not publicly available because they contain sensitive data to be treated under data protection laws and regulations. Appropriate forms of data sharing can be arranged after a reasonable request to the Correspondence authors or the Ethics Review Committee of General Hospital of Central Theater Command of the Chinese People’s Liberation Army.

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
