# Peer review of "Analysis of the Causes of Solitary Pulmonary Nodule Misdiagnosed as Lung Cancer by Using Artificial Intelligence: A Retrospective Study at a Single Center"

_diagnostics, 2022, doi:10.3390/diagnostics12092218_

Round 1

Reviewer 1 Report

Dear authors,

I think that the article is interesting and the topic is surely of clinical interest and still debated. However, some points need to be improved and cleared:

- Line 89-90: the patient selection should be improved, please describe the details. (example, Inclusion/exclusion criteria: has performed a pathological diagnosis within 2 months, etc)

- Line 93-94: please clarify how the radiologists has reached the agreement and the consequent updated statistical analysis

- Line 125-126 please add the total number of patient and the demographic data for the whole sample.

Author Response

- Line 89-90: the patient selection should be improved, please describe the details. (example, Inclusion/exclusion criteria: has performed a pathological diagnosis within 2 months, etc)

Response: We greatly appreciate you for these constructive suggestions, which highly improve the quality of this work. We were also concerned about this issue, and have added the sentences in the revised version.

- Line 93-94: please clarify how the radiologists has reached the agreement and the consequent updated statistical analysis

Response: Thanks for your kind suggestions, which are valuable for improving the accuracy of the manuscript. We were also concerned about this issue, and have added the sentences in the revised version.

- Line 125-126 please add the total number of patient and the demographic data for the whole sample.

Response: Thank you for your patience to carefully review. We have improved it in the revised manuscript.

Reviewer 2 Report

First of all, the title is misleading. Authors present a serie of cases, in fact very limited (21 patients, 61 SPNs), of pulmonary nodules diagnosed in 2020, but there are not other referrals to COVID-19.

Authors have to better describe the populations: 61 SPNs in 21 patients are too much, they have to describe how many nodules have each patient.

Authors wrote that the "All SPNs were confirmed by pathology or followed up with clinical treatment." They have to specify how many nodules had the final histopathological diagnosis and how many are in follow up, and how long was the follow up. In my opinion the f-up it's stilltoo short because the cases were collected between January 2020 and August 2020, so the median follow up it's shorter than 2 years, and it could not be significant.

Lines 61-63: the sentence it's not clear, english must be improved

Line 67: replace "2019(COVID-19)" with "2019 (COVID-19)"

Line 70: the reference [5] cited it's not about the COVID-19 epidemic, as Authors wrote. The work was published in 2017, prior to the COVID-19.

Line 83: the reference [10] it's not about AI as Authors wrote in the text. The percentage reported are not in the work cited.

Line 182: "The maximum diameter of 182 the SPN was 21*30 mm." What do that mean??

Line 184-185: "The average 184 volumeoftheSPNwas3280±2646mm3" How it is possible?? That numbers are huge

All the nodules shown are GG nodules. In Figure legends Authors didn't specify the final diagnosis of those nodules.

Author Response

First of all, the title is misleading. Authors present a serie of cases, in fact very limited (21 patients, 61 SPNs), of pulmonary nodules diagnosed in 2020, but there are not other referrals to COVID-19.

Response: Thank you for your in-depth review and valuable comments, which have helped us tremendously to improve the quality and clarity of the report. Artificial intelligence has been applied to lung cancer imaging diagnosis, which was a very mature technology. This paper only identifies the limitations of AI in a particular period during COVID-19 epidemic. So, the title and content of this study highlight solitary pulmonary nodule misdiagnosed as lung cancer by AI. Common imaging signs of COVID-19 that are not generally misdiagnosed as lung cancer were not included in the study.

Authors have to better describe the populations: 61 SPNs in 21 patients are too much, they have to describe how many nodules have each patient.

Response: Thanks for your kind suggestions, which are valuable for improving the accuracy of the manuscript. We were also concerned about this issue, and have added the sentences in the revised version. A total of 61 patients with 61 SPNs were enrolled in the study, 21 of whom had confirmed COVID-19 by throat swab RT-PCR. Another 40 patients were diagnosed with SPN by postoperative pathological, including 34 malignant SPNs and 6 benign SPNs. So, the final diagnosis was 34 malignant SPNs and 27 benign SPNs (including 21 benign SPNs of COVID-19 and 6 benign SPNs of tumor).

Authors wrote that the "All SPNs were confirmed by pathology or followed up with clinical treatment." They have to specify how many nodules had the final histopathological diagnosis and how many are in follow up, and how long was the follow up. In my opinion the f-up it's stilltoo short because the cases were collected between January 2020 and August 2020, so the median follow up it's shorter than 2 years, and it could not be significant.

Response: We greatly appreciate you for these constructive suggestions, which highly improve the quality of this work. We were also concerned about this issue, and have added the sentences in the revised version. We retrospectively analyzed the clinical data of patients who were detected with SPN for the first time by the chest CT examinations from January 22, 2020, to August 15, 2020. All included patients should have results of postoperative pathological or have the results of RT-PCR positive for COVID-19 within one month.

Lines 61-63: the sentence it's not clear, english must be improved

Response: We felt so sorry for our negligence. We have revised our manuscript accordingly.

Line 67: replace "2019(COVID-19)" with "2019 (COVID-19)"

Response: Thank you for pointing out this lack of preciseness place and guiding us to improve it. We have revised our manuscript accordingly.

Line 70: the reference [5] cited it's not about the COVID-19 epidemic, as Authors wrote. The work was published in 2017, prior to the COVID-19.

Response: We would like to thank you for this helpful advice. I think there's been some misunderstanding, the reference [5] is not about COVID-19, just as “Long-term labor would fatigue the doctor and reduce the effectiveness of screening, leading to a missed diagnosis of SPN”

Line 83: the reference [10] it's not about AI as Authors wrote in the text. The percentage reported are not in the work cited.

Response: Thank you for pointing out this shortcoming. We have revised our manuscript accordingly.

Line 182: "The maximum diameter of 182 the SPN was 21*30 mm." What do that mean??

Response: We felt so sorry for our negligence. We have revised our manuscript accordingly. The size of the SPN was 21×30 mm.

Line 184-185: "The average 184 volumeoftheSPNwas3280±2646mm3" How it is possible?? That numbers are huge

Response: Thank you for posing such nice question. If the SPN size was 13mm×14mm×18mm, the volume of the SPN was 3276 mm3

All the nodules shown are GG nodules. In Figure legends Authors didn't specify the final diagnosis of those nodules.

Response: Thank you for posing such nice suggestion. We have added the diagnosis of those nodules in the revised version.

Reviewer 3 Report

The study is quite relevant and interesting. However, due to the limited sample size, the lack of correlation with pathological findings and the poor English structure and grammar of some sentences throughout the manuscript, the reviewer cannot recommend accepting this work in the present form.

Specific comments:

1.     Line 35, rewrite the sentence. The reviewer finds it very confusing

2.     Lines 36-38, rewrite the sentence. Similarly, Lines 83-86. The reviewer finds it difficult to understand

3.     Line 46, please correct the grammar so it reads “COVID-19 patients with SPN might have been misdiagnosed using the AI system”.

4.     Line 64, correct the grammar in the 1st part of the sentence.

5.     According to the CT scan acquisition description, the CT acquired was not a high resolution scan. This could have affected the detection accuracy of SPN. Please elaborate.

6.     Line 114-115, is there a typo mistake or redundancy? Specifically, in calculating the correct prediction ratio?

7.     What about pathologically confirmed SPNs, shouldn’t it be the ground truth instead of radiologists findings. All the comparisons were made between radiologists and AI findings!

8.     Sample size of 21 cases seems to be too small to draw a solid conclusion.

9.     Lines 303-305 explain the reason behind he limited accuracy of predicting SPNs in this study. Hence what is the message authors are aiming to deliver?

10.  What are the potential limitations encountered in this study?

Author Response

  1. Line 35, rewrite the sentence. The reviewer finds it very confusing.

Response: Thank you for pointing out this shortcoming. We have revised our manuscript accordingly.

  1. Lines 36-38, rewrite the sentence. Similarly, Lines 83-86. The reviewer finds it difficult to understand

Response: Thanks for your kind suggestions, which are valuable for improving the accuracy of the manuscript. We had edited our manuscript with the aid of the AJE website.

  1. Line 46, please correct the grammar so it reads “COVID-19 patients with SPN might have been misdiagnosed using the AI system”.

Response: We agree with the suggestions of you. Thank you for your help.

  1. Line 64, correct the grammar in the 1st part of the sentence.

Response: Thanks for your kind suggestions, which are valuable for improving the accuracy of the manuscript. We had edited our manuscript with the aid of the AJE website.

  1. According to the CT scan acquisition description, the CT acquired was not a high resolution scan. This could have affected the detection accuracy of SPN. Please elaborate.

Response: Thank you for your in-depth review and valuable comments, which have helped us tremendously to improve the quality and clarity of the report. The scanning parameters were as follows: tube voltage of 120 kV, adjusted tube current that ensured that the CTDIvol value was 7 mGy, scanning layer thickness and layer spacing of 0.5–2 mm, spiral pitch of 1.3, and pedal scan direction. Which was a high resolution scan [DOI: 10.1371/journal.pone.0248957].

  1. Line 114-115, is there a typo mistake or redundancy? Specifically, in calculating the correct prediction ratio?

Response: Thanks for your kind suggestions, which are valuable for improving the accuracy of the manuscript. We have revised our manuscript accordingly.

  1. What about pathologically confirmed SPNs, shouldn’t it be the ground truth instead of radiologists findings. All the comparisons were made between radiologists and AI findings!

Response: We greatly appreciate you for these constructive suggestions, which highly improve the quality of this work. We were also concerned about this issue, and have added the sentences in the revised version. We retrospectively analyzed the clinical data of patients who were detected with SPN for the first time by the chest CT examinations from January 22, 2020, to August 15, 2020. All included patients should have results of postoperative pathological or have the results of RT-PCR positive for COVID-19 within one month. We have added the diagnosis of those nodules in the revised version.

  1. Sample size of 21 cases seems to be too small to draw a solid conclusion.

Response: We want to thank the reviewer for pointing out this very fundamental comment. This was a very important to us as well, and we gave it a lot of thought. We have added the paragraph of limitations according to the reviewers’ comments in the revised manuscript.

  1. Lines 303-305 explain the reason behind he limited accuracy of predicting SPNs in this study. Hence what is the message authors are aiming to deliver?

Response: Thank you for posing such nice suggestion. We want to express that the accuracy of AI requires deep learning, and the sample size in this study is not large.

  1. What are the potential limitations encountered in this study?

Response: Thank you very much for your valuable advice, the sample size in this study is not large.

Reviewer 4 Report

The abstract needs quantification. Covid 19 analysis and prediction is as good but the paper lags scientific temper of methodology. AI accuracy is very low why?. The authors may included MCC and kappa analysis for further improvement. Some more classification techniques like SVM classifier may also be studied. The conclusion is poorly presented in the paper. The discussion needs improvement.

Author Response

Response: We want to thank the reviewer for pointing out this very fundamental comment. This was a very important to us as well, and we gave it a lot of thought. We have optimized the limitations according to the reviewers’ comments in the revised manuscript.

Round 2

Reviewer 1 Report

I believe that the authors have improved their submission, which addresses interesting diagnostic results. There are no major scientific deficiencies that would prevent publication.

Author Response

Thank you very much for your positive comments. Your encouragement will be the driving force for us to move forward.

Reviewer 2 Report

You didn't adequately answered to my comments.

My comment were precise and specific, and you answered in a very inaccurate and vague way, as you I reported below (You can find my new comments to your "responses" in red):

1- First of all, the title is misleading. Authors present a serie of cases, in fact very limited (21 patients, 61 SPNs), of pulmonary nodules diagnosed in 2020, but there are not other referrals to COVID-19.

Response: Thank you for your in-depth review and valuable comments, which have helped us tremendously to improve the quality and clarity of the report. Artificial intelligence has been applied to lung cancer imaging diagnosis, which was a very mature technology. The references about it were too old, and didn't justify the "maturity" of the technology as they mentioned them. This paper only identifies the limitations of AI in a particular period during COVID-19 epidemic. So, the title and content of this study highlight solitary pulmonary nodule misdiagnosed as lung cancer by AI. SO you have to delete the name "COVID-19" from the title because in this way it's very misleading. Common imaging signs of COVID-19 that are not generally misdiagnosed as lung cancer were not included in the study. You didn't include neither imaging signs that could misdiagnosed as lung cancer. This response is very vague and inaccurate.

2- Authors have to better describe the populations: 61 SPNs in 21 patients are too much, they have to describe how many nodules have each patient.

Response: Thanks for your kind suggestions, which are valuable for improving the accuracy of the manuscript. We were also concerned about this issue, and have added the sentences in the revised version. A total of 61 patients with 61 SPNs were enrolled in the study, 21 of whom had confirmed COVID-19 by throat swab RT-PCR. Another 40 patients were diagnosed with SPN by postoperative pathological, including 34 malignant SPNs and 6 benign SPNs. So, the final diagnosis was 34 malignant SPNs and 27 benign SPNs (including 21 benign SPNs of COVID-19 and 6 benign SPNs of tumor). As I wrote in my previous revision, you have to specify the histopathological diagnosis of each patient! The 21 patient who had COVID had also an histopathological diagnosis of their SPNs? It's not clear! You didn't answered.

Authors wrote that the "All SPNs were confirmed by pathology or followed up with clinical treatment." They have to specify how many nodules had the final histopathological diagnosis and how many are in follow up, and how long was the follow up. In my opinion the f-up it's stilltoo short because the cases were collected between January 2020 and August 2020, so the median follow up it's shorter than 2 years, and it could not be significant.

Response: We greatly appreciate you for these constructive suggestions, which highly improve the quality of this work. We were also concerned about this issue, and have added the sentences in the revised version. We retrospectively analyzed the clinical data of patients who were detected with SPN for the first time by the chest CT examinations from January 22, 2020, to August 15, 2020. All included patients should have results of postoperative pathological or have the results of RT-PCR positive for COVID-19 within one month. Once again, you didn't answered to my comment. You wrote that some patients had a final histopathological post-surgical diagnosis of their SPN and some others had a follow up of the nodule, but you didn't mentioned how long was this follow up, and it's sure that this would be shorter than 2 years, so it could not be considered significant. It's clear that I didn't meant the COVID-19 follow up. I think that your answers are captious.

Author Response

Dear Prof. Dr. Andreas Kjaer, Editors, and reviewers,
Thank you very much for the opportunity to resubmit our revised manuscript. We are very appreciative of these constructive suggestions and comments. In the last revision process, our answer was not satisfactory to the reviewer 2 in last revising due to language barriers. We are deeply sorry. Here, we sincerely apologize to all teachers of reviewer. I sincerely request the teacher accept our sincere apology.  All the problems you pointed out are helping us optimize the manuscript, and we are very grateful to you. Thank you very much for the guidance of teacher. Finally, please allow us to deeply apologize.
Our point-by-point answers to the reviewers’ comments are below, with amendments highlighted in blue.

Reviewer #2:
Comment 1: First of all, the title is misleading. Authors present a serie of cases, in fact very limited (21 patients, 61 SPNs), of pulmonary nodules diagnosed in 2020, but there are not other referrals to COVID-19. This paper only identifies the limitations of AI in a particular period during COVID-19 epidemic. So, the title and content of this study highlight solitary pulmonary nodule misdiagnosed as lung cancer by AI. SO you have to delete the name "COVID-19" from the title because in this way it's very misleading. Common imaging signs of COVID-19 that are not generally misdiagnosed as lung cancer were not included in the study. You didn't include neither imaging signs that could misdiagnosed as lung cancer. 
Response: Thank you very much for your valuable advice. We have deleted the "COVID-19" from our manuscript title. Since the epidemic of COVID-19 in Wuhan lasted only a few months, so there are not patients with malignant solitary pulmonary nodules referrals to COVID-19 in our hospital. Common imaging signs of COVID-19, which was not a solitary pulmonary nodule, are not generally misdiagnosed as lung cancer were not included in the study. 

Comment 2: As I wrote in my previous revision, you have to specify the histopathological diagnosis of each patient! The 21 patient who had COVID had also an histopathological diagnosis of their SPNs? It's not clear! You didn't answered. 
Response: We apologize for our negligence. Patients were followed for one month if COVID-19 was confirmed. The pulmonary nodules disappeared in all of these patients during follow-up, and no pathological examination results were available in all of them. So, a total of 61 patients were enrolled in the study, 21 of which had confirmed COVID-19 by throat swab RT‒PCR. The pulmonary nodules disappeared in all of these patients during follow-up, and no pathological examination results were available in all of them. The remaining 40 patients were diagnosed with SPNs by postoperative pathology, including 34 malignant SPNs and 6 benign SPNs. 

Comment 3: Once again, you didn't answered to my comment. You wrote that some patients had a final histopathological post-surgical diagnosis of their SPN and some others had a follow up of the nodule, but you didn't mentioned how long was this follow up, and it's sure that this would be shorter than 2 years, so it could not be considered significant. It's clear that I didn't meant the COVID-19 follow up. I think that your answers are captious. 
Response: Thank you for pointing out this shortcoming. Patients were followed for one month if COVID-19 was confirmed. The pulmonary nodules disappeared in all of these patients during one-month follow-up. In our opinion, pulmonary nodules in COVID-19 patients will disappear in a short time after active treatment, and it is not necessary to obtain pathological examination through surgery. We are sorry for not explaining this clearly in the draft. We have added the sentences in the revised draft.

Finally, we again thank the editors and the reviewers for your great efforts in improving the quality of this manuscript.
We look forward to hearing from you regarding our submission. We would be happy to respond to any further questions and comments that you may have.
 Thank you and best regards.

Reviewer 3 Report

Specific comments:

Line 96-98, the sentence starting with This study analyzing is redundant, should be deleted.

Line 130, 21 of which instead of 21 of whom.

Line 131, the remaining 40 instead of another 40.

Author Response

Response: Thank you for your careful review. We have improved the three errors in the revised manuscript.

Reviewer 4 Report

The following questions are not answered in the revised paper AI accuracy is very low why?. The authors may included MCC and kappa analysis for further improvement. Conclusion /summary has to be added in the paper.

Author Response

Response: Thank you for pointing out this shortcoming. We used a paragraph discussing why AI accuracy is very low (line 274 to 294).

Thank you for your patient advice. We have included MCC and kappa analysis in the revised manuscript (table 1.3). And we added MCC and kappa analysis results in abstract. Summary paragraph had added in revised manuscript.

Round 3

Reviewer 2 Report

Authors changed the manuscript following my previous comments.